# Age and origin of a Cahokian wooden monument at the Mitchell site, Illinois, USA

**Nicholas V. Kessler**[1]*, **Erin M. Benson**[2°], **Timothy R. Pauketat**[2°], **Jason D. Kirk**[3], **Matthew D. Therrell**[4]

**1** Laboratory of Tree-Ring Research, University of Arizona, Tucson, United States of America, **2** Illinois State Archaeological Survey, University of Illinois, Urbana, United States of America, **3** Department of Geosciences, University of Arizona, Tucson, United States of America, **4** Department of Geography, University of Alabama, Tuscaloosa, Alabama, United States of America

☉ These authors contributed equally to this work

* nvkessler@arizona.edu

## Abstract

Cahokia was the first and largest precolonial city outside of Mesoamerica in what is now the United States. Monuments and exotic goods were central to public life at Cahokia, but no high-resolution timeline of monumental construction or long-distance material import exists for the site. Wooden marker posts, serving as both public monuments and exotic artifacts, offer ideal sources of evidence for documenting the chronology and spatial scale of Cahokian material networks and community histories. In this paper, we employ $^{14}$C dating of a cosmic event archived in tree-rings to determine that the largest known marker post in the Cahokia area, the Mitchell Log, was felled around 1124 CE. Sr isotope ratios of the wood rule out a local source, and suggest the tree was transported at least 180 kilometers. Together, the date, provenance, and context of the Mitchell Log (1) establish a historical datum for the peak influence of the Cahokia polity, (2) prompt new questions about the long-distance transport of thousands of other such marker posts, and (3) identify a significant event in the history of this precolonial phenomenon.

## Introduction

Cahokia emerged spectacularly as the largest city north of modern-day Mexico around 1050 CE [1,2]. With a population of 20,000 or more, Cahokia consisted of three contiguous precincts sprawled across the central portions of the northern "American Bottom," a historically named expanse of Mississippi River floodplain within an even larger "Greater Cahokia" region [3–6]. By around 1100 CE, Cahokia was the preeminent political and administrative center in the "Mississippian" midcontinent and Southeast, but its population and influence began to decline before 1200 CE [7–9]. The rate and nature of this decline is uncertain, with hypotheses focused on climate change, resource depletion, environmental catastrophe, and socio-political

**Data availability statement:** All relevant data are within the manuscript and its Supporting Information files.

**Funding:** Illinois Department of Transportation. The funders had no role in study design, data collection and analysis, decision to publish, or preparation of the manuscript. There is no place to state this in the Manuscript Data section so I clarify that here.

**Competing interests:** The authors have declared that no competing interests exist.

factionalism and fragmentation [8–14]. Much archaeological research has focused on reconstructing population and environmental trends in the American Bottom, but the precise timing of major religious and political changes in and around the city is an equally important piece of information [8]. Cahokia's political influence, at its height, was closely linked to the procurement of powerful, often exotic, goods and the regular construction of monuments [2,15]. Understanding the chronology of exotic material imports and changes in monumental construction can help clarify the timing of the peak and denouement of Cahokia's political integration and interregional influence [8].

One important and common type of monumental construction at Mississippian sites are marker posts. Marker posts were large wooden monuments hewn from tree trunks and set vertically in the ground near prominent buildings, atop pyramidal mounds, or in community courtyards, squares, or plazas. The massive poles were moved from time to time, probably in order to mark a precinct's axis mundi, embody a spirit being, or serve as the focal point in pole-climbing or pole-flying rituals [16–20]. Besides occasional pieces of preserved wood, the archaeological evidence for such posts takes the form of deep cylindrical pits with one or more insertion and extraction ramps extending off to the sides.

In the Greater Cahokia region, these large posts were restricted to precincts, towns, nodal sites, and shrines and were unknown until the beginning of Cahokia's urban Lohmann (1050–1100 CE) and Stirling (1100–1200 CE) phases [21]. Researchers have identified hundreds of large, ramped post pits dating to the Lohmann and Stirling phases, the basis for projections that thousands of such poles once stood in and around the central city precincts at any one time between 1050 and 1200 CE (see Emerson et al. 2018). Perhaps the upscaled versions of small Terminal Late Woodland antecedents evident in pre-urban village courtyards [18–20,22], the large Cahokian marker posts appeared with the beginning of the urban Lohmann phase as part of the region's rapid, widespread urbanizing developments. From there, cultural practices involving large marker posts spread across the Midwest, Southeast, and Great Plains after 1100 CE [23], along with evidence of more widespread Cahokian interaction. But Cahokian post emplacements ended sometime in the late twelfth or early thirteenth century CE, as Cahokian urbanism was dissolving. Better chronometric dates for the cessation of marker post acquisition and emplacement should provide historical insights into how this dissolution occurred.

Dendrochronological analysis of cosmic radiation events archived in the $^{14}$C content of tree-rings can provide an absolute chronological reference for wooden objects. In the present study, high-precision tree-ring-radiocarbon dating was used to establish the cutting date of the so-called Mitchell Log (Fig 1), a plaza marker post that we demonstrate was an exotic (to the Cahokia area) bald cypress tree (*Taxodium distichum*) transported to the Cahokian outlier precinct known as Mitchell, a civic-ceremonial site featuring a dozen mounds around a 5-ha plaza 10 km north of the Cahokia precinct [25]. While marker posts were once fairly common in the Cahokian landscape, they are rarely preserved and have never been subject to

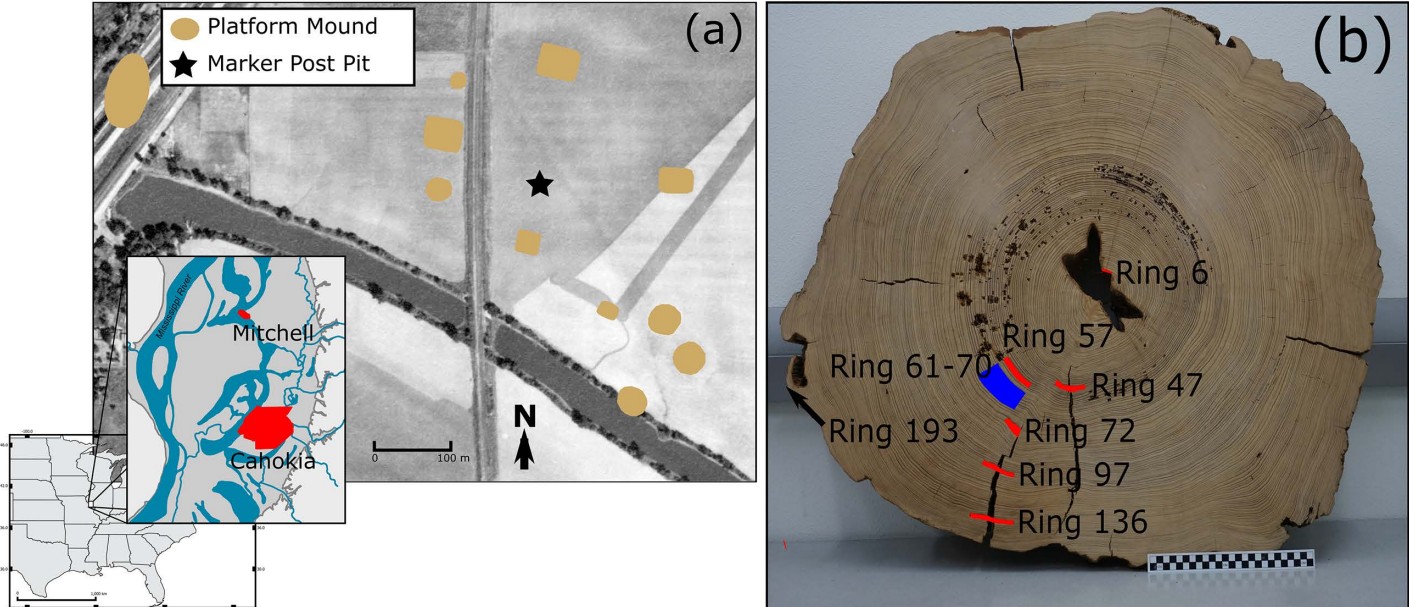

**Fig 1. Mitchell Log Location and Samples.** Location of the Mitchell Log within the site and within Great Cahokia (panel A). Base map is from the "sptools" library in R. Aerial photograph is archival imagery from the Illinois State Geological Survey [24]. Illustration of the location of the $^{14}$C samples in the Mitchell Log ring sequence (panel B).

high-precision numerical dating and sourcing to this point. Therefore, this analysis presents a rare glimpse at the chronology and procurement of this important artifact type.

## Mitchell log chronology

The Mitchell Log is a 3.5-m-long basal remnant of an upright marker post. It was discovered in 1961 in the site's central plaza. The post remnant was abandoned in a ramp of the 4-meter-deep post pit after it had apparently been broken off during the extraction process. Dry, this basal portion of log weighed just over a metric ton and based on the depth of the post pit and the length of the extraction ramp, the marker post may have been at least 18 m tall originally, meaning that the entire pole may have weighed four or five metric tons [26].

Four early radiocarbon dates from the inner, middle, and outer rings of the Mitchell Log yield an estimated date between 1050–1252 CE (95.4% probability), which is broadly consistent with radiocarbon dates obtained from mound contexts [26] (S1). Our goal was to use current tree-ring-radiocarbon (or wiggle-match) calibration techniques to date the ring sequence of the Mitchell Log. Tree-ring-radiocarbon dating is a $^{14}$C calibration method used to date tree-ring series where the number of years between sequential $^{14}$C measurements establish Bayesian priors for calculating the posterior density distributions of calendar dates [27,28]. The $^{14}$C content from annual tree-rings will tend to correspond to the historical variability of atmospheric $^{14}$C, resulting in more precise posterior dates than single determinations (independent of any other Bayesian modeling).

We suspected that the ring series of the Mitchell Log spanned the 993/4 solar energetic particle event (SEP), and the initial tree-ring-radiocarbon model confirmed this. SEPs are one class of solar and galactic events linked to increased atmospheric nuclide production outside the typical 2‰ range of annual variation [29,30]. These short-lived events cause recognizable $^{14}$C enrichments that can be dated to the year in some cases [31–33]. In this study, an annual series of $^{14}$C dates straddling the estimated position of the 994-ring provided additional precision and security for the absolute dates determined for the Mitchell Log.

## Mitchell log provenance

At the time the Mitchell log was excavated, it was one of the few examples of bald cypress identified in the Greater Cahokia region. Since then, more bald cypress wood has been recovered, nearly exclusively from the Cahokia and East St. Louis precincts [34–42]. The earliest known bald cypress wood from Cahokian contexts was recovered from the western periphery of the Cahokia precinct and was dated indirectly using diagnostic pottery to the decade or two before 1050 CE [37,39,40]. All of the rest of the known cypress wood dates to the urban period of Cahokia history (1050–1200 CE), with none known from the later times (1200–1300 CE). At the Mitchell site, cypress is absent from structural wood collections from residential features [43] and given that no archaeological cypress wood is known from outside Cahokia's central precincts, access seems to have been restricted. Bald cypress, and other conifers such as red cedar (Juniperus virginiana) and pine (Pinus spp.) are interpreted as having been restricted to the houses of elite families, special shrines, mortuary areas, and other public monuments based on the distribution of these woods in the central precinct of Cahokia [41,43].

Some researchers suggested that there must have been a local source for the wood [39]. Other scholars argue, on the basis of the contemporary and historical distribution of bald cypress, that it was unlikely that any significant quantities were available around Cahokia or Mitchell but that it was obtained from downriver sources [43]. United States Forest Service stand data [44] and the distribution of International Tree-ring Data Bank study sites [45] indicate that southern Illinois and the lower Ohio River valley represent the approximate northern natural extent of bald cypress today, while historical records indicate that cypress did not extend significantly north of approximately 37°N latitude. Neither is there any reported palaeoecological evidence to support local American Bottom bald cypress sources [43]. There are, however, infrequent discoveries of southern yellow pine (likely shortleaf pine (Pinus echinata)) wood at or around Cahokia, a group which likewise has a natural range that extends no closer to Cahokia than 120 km to the southwest [11,46,47]. Pine is also found uncarbonized, and sometimes shaped, indicating that it, like cypress, was not primarily used as fuel and may have been imported for special architectural and monumental purposes [41].

Sr isotope ratios ($^{87}$Sr/$^{86}$Sr) from the outer heartwood rings of the Mitchell Log were used to evaluate the relative probability of a local source for the Mitchell Log. Sr is assimilated by trees through soil water and should closely match the Sr of bedrock, soil, and animals in the local environment [48]. In the midcontinental U.S., geologic variability is sufficient to allow the comparison of Sr values to once-living plants and animals to gridded Sr values from the environment to determine provenance and movement patterns [49–51]. Sr sourcing of archaeological wood is complex. Multiple studies point to the incorporation of atmospheric Sr in trees as a complicating factor for distinguishing specific source areas [52–54]. Despite this, studies report success in using Sr values of archaeological wood to rule out local source areas [54,55]. Recently, in the Midwestern U.S., researchers were able to discriminate local from extra-local sources of a wooden bowl using reference values from Sr isoscapes at the regional scale [49]. Importantly, these studies uniformly suggest that Sr values in wood are not affected by diagenetic processes that can obscure signal, and that Sr is a valuable tool for wood sourcing at gross scales.

## Materials and methods

### Dendrochronology and $^{14}$C dating

Dendrochronological analysis focused on an intact cross-section curated at the Illinois State Archaeological Survey for the Illinois Department of Transportation. No permits were required for the described study, which complied with all relevant regulations. Goals for this analysis were to measure ring widths to confirm internal cross-matching and ring counts as well as to attempt crossdating with reference chronologies. Ring width measurements were cross-matched and compared to existing long tree-ring chronologies from the region including the Allred Lake and Cache River chronologies [56,57] and a sub-fossil oak chronology from northern Missouri [58].

Rings were examined along multiple radii aided by a dissecting microscope. Because of the size of specimen, ring widths were measured with a digital caliper at 0.01 mm precision. Fragments of a replicate specimen were measured previously by one of the co-authors (MDT). Cross matching with the ring width measurements was conducted using the COFECHA program and the dplR package in the R computer program [59,60]. Sapwood is preserved on the specimen, as is an apparent bark ring (waney edge) with tool marks (S1). Those familiar with the wood collections and their curation also observed bark adhering to other pieces of the log archived in collections at Southern Illinois University, Carbondale.

Initial, low-resolution, $^{14}$C sampling consisted of individual rings and aggregates of up to 5 rings (average 2 rings) separated by between 10 and 39 years (average 20-year gap). Approximately 70−200 mg of raw wood was sampled, and initial weights were recorded. Pretreatment involved Soxhlet washes in a series of solvents (acetone – hexanes – ethanol – methanol – water) designed to remove common oil-based and acrylic conservation treatments from wood [61,62]. Standard ABA treatment followed to remove inorganic carbonates and humic substances following [63]. Samples were rinsed in Type-1 water to a neutral pH and decomposed to holocellulose in sodium chloride bleach, after which the samples were rinsed to a neutral pH and alpha cellulose was extracted with 17.5% sodium hydroxide solution [64]. Following final neutralization, the samples were dried, and post treatment weights were recorded. Sample combustion, isotopic analysis, and graphitization was carried out using an in-line system consisting of a Vario ISOTOPE Select element analyzer coupled with an Elementar Isoprime PrecisionIon mass spectrometer and an IonPlus automated graphitization equipment. $^{14}$C measurements were obtained at the Center for Applied Isotopic Studies at the University of Georgia and the Carbone Lab at Northern Arizona University.

Radiocarbon dates were calibrated against the IntCal20 curves [65] using the OxCal 4.4 software package [66] and wiggle-matching D_Sequence() models based upon Bronk Ramsey et al. [27]. For annual dating against the 993/4 SEP, measured $^{14}$C concentrations of annual tree-ring samples were matched to all annual reference values from IntCal20 [67–75]. The $\chi 2$ method ($\alpha = 0.05$) was employed for matching the annual $^{14}$C measurements of the Mitchell log to the annual reference data [33].

Additional $^{14}$C measurements were obtained from 10 annual rings in the region of the ring segment estimated to span the 993/4 SEP based on the initial wiggle-match. Replicate measurements were obtained from the rings thought to date to 994, surrounding years, as well as the beginning and end of the annual segment for a total of 15 measurements (S1). The replicates were measured from the same pretreated material and so their pooled mean $^{14}$C age and standard deviation was calculated with an added systematic uncertainty of 1‰. The OxCal CQL script for reproducing the initial D_Sequence model is provided in S2 and the R script for reproducing the annual tree-ring-radiocarbon calibration is provided in S3.

## Sr sourcing

Strontium ($^{87/86}$Sr) values in the Mitchell Log were measured at the Thermal Ionization Mass Spectrometry facility, University of Arizona. Approximately 0.15 g to 0.5 g of wood was combusted in glass vials in a muffle furnace at 650 °C for 15 hours. The resulting wood ash was dissolved in 2 ml of 8N nitric acid and 1 ml of hydrogen peroxide, and the solution was dried in a HEPA-filtered dry box overnight. The residue was re-dissolved in 2 ml of 8N nitric acid. Strontium was separated and purified on polypropylene disposable columns (Bio-Spin® Chromatography Columns, Bio-Rad, Hercules, California, part number 7326008) using Sr resin (Sr-Spec Resin®) and twice-distilled acids [76]. Purified Sr was loaded unto Tantalum (Ta) filaments with a TaF5 activator [77]. A VG Sector 54 multi-collector thermal ionization mass spectrometer in dynamic collection mode was used for the Sr isotopic analyses. The 86Sr/88Sr value of 0.1194 was utilized to correct the 87Sr/86Sr ratios for mass-dependent fractionation. The average NIST SRM 987 standard value during the sample runs was 0.710265 ± 0.000009 (n = 3, 1s). Total procedural blanks were approximately 100 pg, and an inconsequential proportion of the total strontium was separated from each sample.

The analytical approach was to compare measured Sr isotopes of the Mitchell Log to gridded reference values and assess the relative likelihood of a local, American Bottom, source without necessarily determining a specific source area

of growth. A posterior probability surface (isoscape) was calculated to compare Sr measurements to identify the probability of different source areas for the Mitchell Log. Reference values were the global gridded weathered bedrock Sr dataset accessed through waterisotopes.org [78]. Using weathered bedrock values reflects differential weathering process and Sr inputs that can alter bioavailable Sr in the environment, while also representing a more local signal than water which can integrate Sr from a larger spatial extent [48]. This data set is uncalibrated, and this analysis assumes that the isoscape provides a good proxy for local trees. Calculation of the posterior probability surface was conducted using the functions provided by the assignR package in R [79]. The odds ratio was calculated as:

$$\frac{P_{Si}}{P_{AB}}$$

(1)

Where $P_{Si}$ is the mean posterior probability of all isoscape grid cells overlapping each ith polygon estimating the spatial extent of bald cypress digitized from National Forest Service stand data [44]. $P_{AB}$ is the mean posterior probability of the isoscape within the American Bottom. The R-script for reproducing the Sr sourcing analysis is provided in S4.

Results were compared to Mississippian-aged faunal values from across the region to verify the results of the isoscape [50]. The comparison between faunal and tree Sr values is valid because Sr isotopes are not fractionated from bedrock reservoirs across food webs, and there is an expected strong correlation between substrate Sr values and those of plants and animals [80].

## Results

### Dendrochronology and $^{14}$C dating

Our examination confirmed the species identification of bald cypress and located the probable terminal ring (S1). We measured five ring-width series spanning 194 extant rings from the Mitchell Log. Four series were measured from fragments archived at Southern Illinois University, Carbondale and one from a complete cross section (Fig 1) archived in the Illinois Department of Transportation collections at the Illinois State Archaeological Survey, Champaign, and transported to the University of Arizona for analysis. The measurements are significantly correlated and confirm the ring counts used in the $^{14}$C calibration. The combined measurement series have an interseries correlation of 0.74, with all lagged 50-year segments above the critical correlation threshold of 0.32 at 99% probability in the COFECHA program. Cross-dating the Mitchell Log ring-width segment with regional long bald cypress and oak chronologies showed no coherent cross-date. This could be due to a lack of common signal, or more likely, due to lack of temporal overlap (e.g., the Mitchell Log died/was cut before 1185). The outermost ring is incomplete (early-earlywood), indicating tree death before the formation of late wood in summer.

A total of 22 $^{14}$C dates were obtained from the Mitchell Log (S1), and the sequence was wiggle-matched in two stages. Uncalibrated dates for the initial wiggle-match are summarized in S1 Table 1. Calibrating the D_Sequence() of seven $^{14}$C measurements against IntCal20 resulted in high overall agreement (Acomb = 145) with excellent visual agreement with the calibration data at their modeled dating positions (Fig 2). The age of the outermost ring in the D_Sequence() dates to 1122 ± 4 (1σ) or 1113–1133 (95.4% posterior density).

The visual match of the annually sampled segment suggests that cambial age ring 66 corresponds to the year 994, and this placement has the best fit and suggests a cutting date of 1124 ($\chi 2$ = 4.6, critical value with 10 degrees of freedom = 18.3). The $\chi 2$ values for surrounding years are also significant at the 0.05 level (Fig 3), and thus, cutting dates ranging from 1122–1126 and 1129–1132 cannot be ruled out statistically.

### Sr sourcing

The two Mitchell log Sr measurements returned values of 0.71178 and 0.71173. Examining the resulting isoscape in Fig 4 shows that the areas of highest probability for the Sr match are in what is now southern Missouri, northern Arkansas,

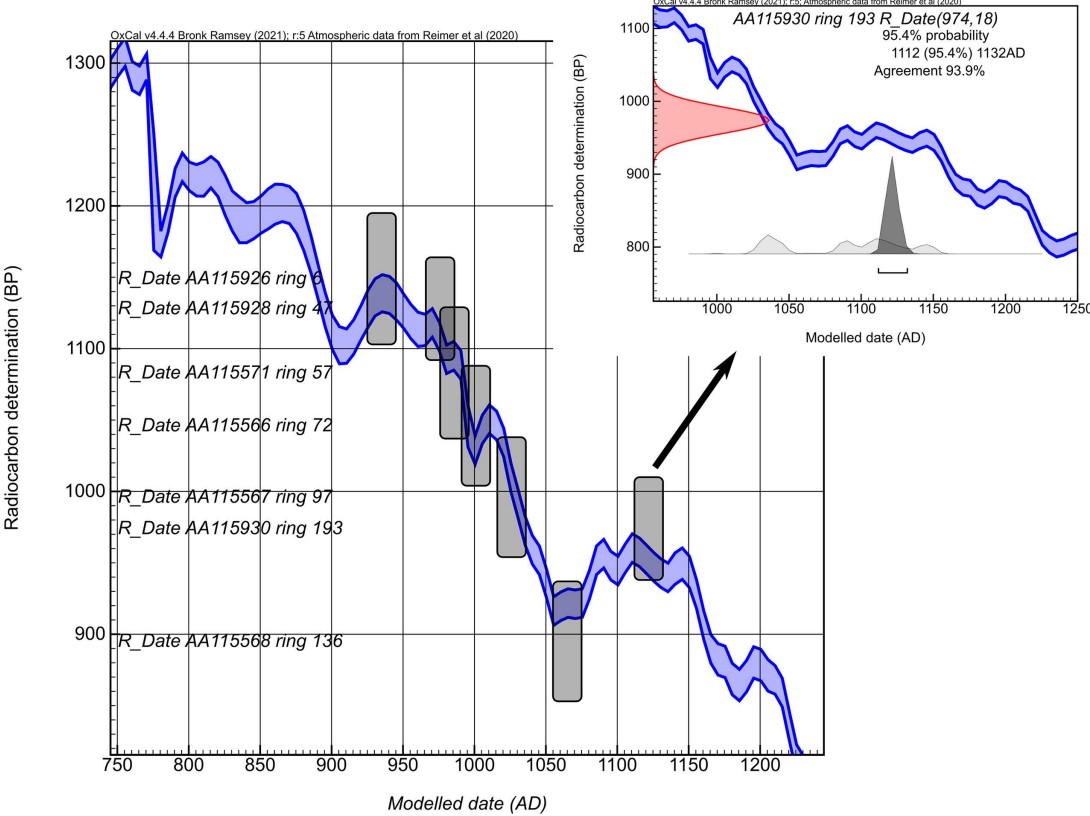

**Fig 2. Initial wiggle-match model.** Measured ¹⁴C content from the Mitchell Log calibrated (D_sequence) against IntCal20 (Reimer et al. 2020). The bars indicate the 2σ ranges of ¹⁴C age (vertical) and calibrated age (horizontal). Inset illustrates the calibrated posterior distribution of the outermost measured ring (ring 193). The outermost extant ring (cutting date) is ring 194.

western Tennessee, and relatively small areas in southern Illinois. Expansive areas of moderate probability are observed across the northern Mississippi River Embayment.

Calculating the area adjusted odds ratio (OR) for contemporary bald cypress stands compared to the American Bottom helps assess the relative likelihood of each non-local source area (Fig 4). While the OR of the different potential source areas is not decisive, it does provide a ranking of possible provenances and can help rule out unlikely alternatives. Adopting 3x OR as a heuristic threshold for substantial support of a hypothesis [83], several potential source areas are compelling alternatives to the American Bottom. The nearest contemporary cypress stands above this threshold are in southern Illinois, along the Ohio River. Additionally, the west-flowing tributaries of the Mississippi River in western Tennessee and modern cypress stands in northeastern Arkansas stand out as potential source areas with OR >100.

Comparison with published faunal Sr measurements reinforces the argument that the Mitchell Log was obtained outside the American Bottom. As observed by Hedman et al. [50], despite the complexity of the geology of the region, American Bottom Sr values are distinguishable from locations farther south. The Mitchell Log Sr measurements (~0.7118) are outside the range of values previously reported for the American Bottom and is not compatible with the "Cahokia signature" (0.7089–0.7097) of Sr isotopes from bone reported by others (Slater et al. 2014). In fact, the only faunal Sr values from the area of modern cypress distribution overlapping the Mitchell Log at 2σ in the Hedman et al. [50] dataset are from the lower Ohio River Valley.

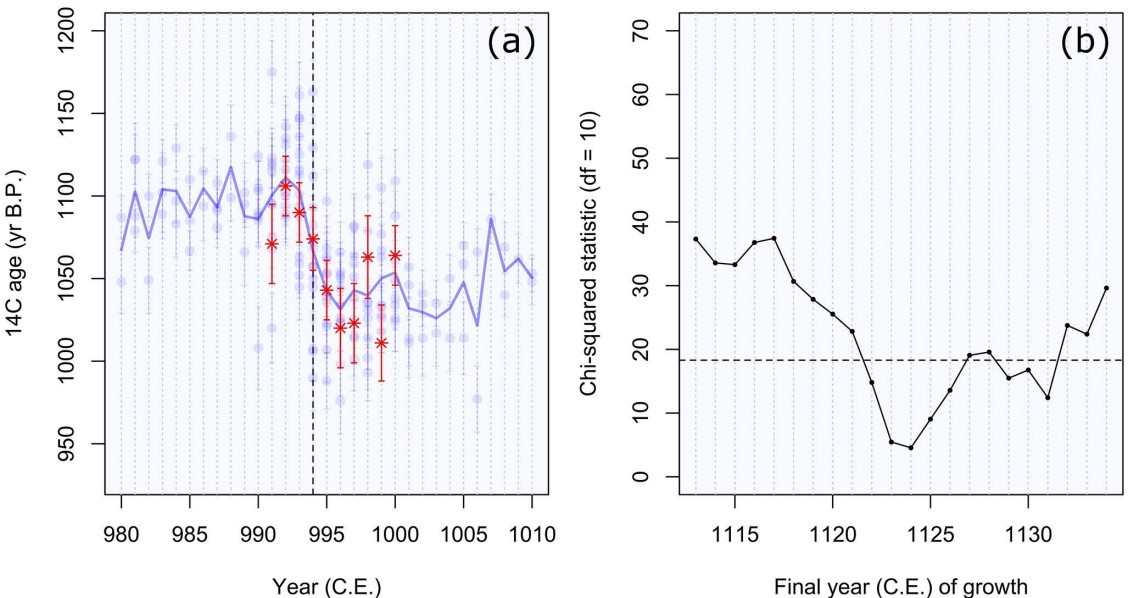

**Fig 3. SEP date.** Comparison of the measurement series (red asterisk) with annual reference values (blue dots) (a). In panel (a) the bars around each point represent the 1σ error on the ¹⁴C measurement, and the blue line corresponds to the mean of the reference values, and the vertical dashed line corresponds to the year 994. Measurement values are plotted in their position that corresponds to the highest χ2 value. Panel (b) displays the χ2 score for each calendar year placement for the final year of tree growth.

## Discussion

As objects that are both monuments and exotic artifacts, potentially datable to the year, wooden marker posts are ideal artifacts with which to grapple with the questions about the nature and scale of the Cahokian urban phenomenon. Minimally, the Mitchell Log provenance and cutting date suggests that Cahokian networks extended far to the south during Cahokia's Stirling-phase peak.

As a point of comparison, the builders of the contemporaneous cultural center of Chaco Canyon in the American Southwest famously imported large quantities of timber from distant sources [84,85]. Around 70% of construction beams for Chaco great-houses were sourced from mountains approximately 75 km away, while a minority of timbers came from as far as 140 km, and the timing of shifts in timber procurement mirrored reorganization in other material networks such as pottery as well [85,86]. In the Greater Cahokia region, the nearest contemporary bald cypress stands with reasonable odds as a source area based on Sr ratios is southern Illinois, which is around 180 km (linear) from the Mitchell site, and provides a minimum figure for the distances involved in obtaining large bald cypress marker posts. Other sources in southern Illinois are more than 200 km distance from Mitchell, while western Tennessee sources could be as far away as 350 km. An even more distant origin for the Mitchell Log is remotely possible, since the isoscape indicates similarities between its Sr measurements and reference values for the Gulf Coast Plain in southern Arkansas and northern Louisiana as well as parts of western Mississippi.

The non-local source area for the Mitchell Log situates marker posts within regional material networks of similar scale involving shell, minerals, and figurines [87,88]. Stirling phase wooden public architecture, including marker posts and circular constructions known as "woodhenges," are linked to the way public rituals and specialized knowledge were leveraged by social or kin groups and their political or religious leaders [16,89,90]. In essence, marker posts of non-local wood types joined other exotic items as symbols of power and authority in Cahokian society. The dating of the Mitchell Log provides one historical datum of this process.

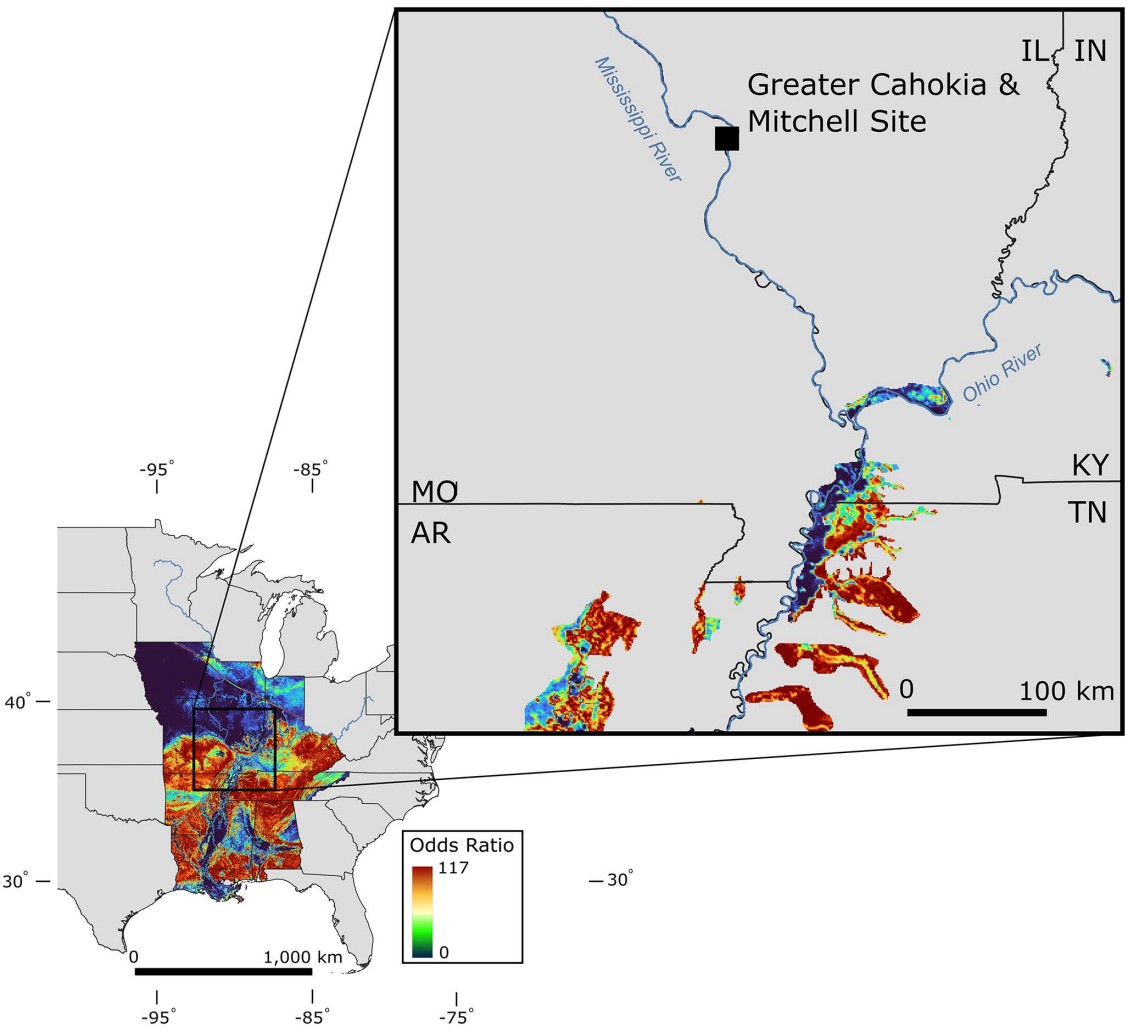

**Fig 4. Mitchell Log isoscape.** Odds ratios of the Mitchell Log Sr source based on the isoscape in the middle and lower Mississippi River Basin (inset) [81] and the odds ratios of an Sr source restricted to modern-day cypress extent and >3 odds ratio in the area nearest the Mitchell Site (expanded panel). Fig 4 was produced in Q_GIS from the data generated in R following procedures described in the text and in S4. Basemap (boundaries and rivers) are from U.S. Geological Survey GIS Data Download [80,82].

The 1122–1126 (or 1129–1132) CE cutting date suggests that the Mitchell Log was procured at the height of Cahokian influence across the Midwest and South, corresponding to the construction of other major monumental structures in the Greater Cahokia region. The timing of the Mitchell Log procurement is within the time interval when earthen and wooden monumental construction had reached a peak, when the regional agricultural system was most highly integrated, and when exotic materials are most in evidence in and around Cahokia's precincts [2,16,91,92]. This suggests that Cahokian logistical capabilities—their ability to organize labor for the farming and central constructions—corresponds to Cahokia's maximum influence up and down the Mississippi River.

The cutting date for the Mitchell Log also informs our understanding of the chronology of monumentality in the Greater Cahokia region. To wit, if the Mitchell Log marker post stood for sufficient time to induce rotting and weakening of the post at the ground level, one or two human generations, then it would have broken in the ground at or shortly after the middle twelfth century. Such an event corresponds to the indirect dating of the end of the use lives of a suite of specialized

Mississippian buildings fronting the Mitchell plaza to the south of the post pit, indicating a terminus ante quem date for their abandonment between about 1150–1175 CE, based on diagnostic ceramics and stratigraphy [6,93–95]. The inferred timing for the removal of the marker post, synchronous with other large-scale modifications of the Mitchell site landscape, also corresponds in time with the beginning of a region-wide political and economic contraction that saw, among other things, the termination of a large portion of the East St. Louis precinct and the abandonment of an upland farming district east of Cahokia, both of which are dated to 1150–1175 CE [19,40,96–98]. Such a timeframe also coincides broadly with a droughty climate, changes in the kinds and quantities of exotic materials in circulation, and a general transformation in the use of public spaces and construction of mounds [21,99–102]. Whether all of Cahokia's marker posts were extracted at the beginning of this regional contraction and transformation remains a question for future researchers. In any case, by 1200 CE, significant regional depopulation and political, religious, and other organizational shifts had been completed and it appears no more oversized marker posts were being emplaced in the Greater Cahokia region [8,13–15,89,97–104].

## Supporting information

**S1. Photograph of outermost extant rings of the dated specimen and all 14C dates reported in the text.**
(DOCX)

**S2. D Sequence model for the initial 14C dating of the Mitchell Log.**
(DOCX)

**S3. R script for reproducing annual tree-ring-radiocarbon calibration.**
(DOCX)

**S4. R script for Sr source analysis.**
(DOCX)

## Acknowledgments

Brad Koldehoff authorized this and the larger reexamination of the original 1960s IDOT archaeology at Mitchell. Pulari Kartha assisted in laboratory pretreatment for the initial round of [14]C measurements. Many thanks to Lucas Wacker who graciously assisted with the wiggle-match of the annual ring segment. Greg Hodgins is thanked for his support in the early stages of this research. Neal Lopinot generously provided information on their earlier studies of the Mitchell Log. Charolette Pearson and David Frank are heartily thanked for their reviews and thoughtful comments on drafts of this manuscript.

## Author contributions

**Conceptualization:** Nicholas V. Kessler.

**Data curation:** Nicholas V. Kessler, Matthew D. Therrell.

**Formal analysis:** Nicholas V. Kessler, Matthew D. Therrell.

**Funding acquisition:** Timothy R. Pauketat.

**Investigation:** Nicholas V. Kessler.

**Methodology:** Nicholas V. Kessler, Jason D. Kirk.

**Project administration:** Nicholas V. Kessler.

**Resources:** Nicholas V. Kessler, Erin M. Benson.

**Supervision:** Nicholas V. Kessler.

**Visualization:** Nicholas V. Kessler.

**Writing – original draft:** Nicholas V. Kessler, Erin M. Benson, Timothy R. Pauketat, Jason D. Kirk, Matthew D. Therrell.

**Writing – review & editing:** Nicholas V. Kessler, Erin M. Benson, Timothy R. Pauketat.

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
