## [Decision Letter · Decision Letter 0]

6 Jul 2025

PONE-D-25-11477Age and origin of a Cahokian Wooden Monument at the Mitchell Site, Illinois, USAPLOS ONE

Dear Dr. Kessler,

Thank you for submitting your manuscript to PLOS ONE. After careful consideration, we feel that it has merit but does not fully meet PLOS ONE’s publication criteria as it currently stands. Therefore, we invite you to submit a revised version of the manuscript that addresses the points raised during the review process.

Please address the reviewer's comment before re-submission.

We look forward to receiving your revised manuscript.

Kind regards,

Peter F. Biehl, PhD

Academic Editor

PLOS ONE

Journal Requirements:

2. In your manuscript, please provide additional information regarding the specimens used in your study. Ensure that you have reported human remain specimen numbers and complete repository information, including museum name and geographic location.

For more information on PLOS ONE's requirements for paleontology and archeology research, see https://journals.plos.org/plosone/s/submission-guidelines#loc-paleontology-and-archaeology-research .

“Illinois Department of Transportation”

5. We note that Figures 1 and 4 in your submission contain map/satellite images which may be copyrighted. All PLOS content is published under the Creative Commons Attribution License (CC BY 4.0), which means that the manuscript, images, and Supporting Information files will be freely available online, and any third party is permitted to access, download, copy, distribute, and use these materials in any way, even commercially, with proper attribution. For these reasons, we cannot publish previously copyrighted maps or satellite images created using proprietary data, such as Google software (Google Maps, Street View, and Earth). For more information, see our copyright guidelines: http://journals.plos.org/plosone/s/licenses-and-copyright.

a. You may seek permission from the original copyright holder of Figures 1 and 4 to publish the content specifically under the CC BY 4.0 license.

Additional Editor Comments:

Please address the reviewer's comment about the post’s attempted extraction.

Reviewers' comments:

Reviewer's Responses to Questions

**Comments to the Author**

1. Is the manuscript technically sound, and do the data support the conclusions?

Reviewer #1: Yes

2. Has the statistical analysis been performed appropriately and rigorously? 

Reviewer #1: Yes

3. Have the authors made all data underlying the findings in their manuscript fully available?

Reviewer #1: Yes

4. Is the manuscript presented in an intelligible fashion and written in standard English?

Reviewer #1: Yes

5. Review Comments to the Author

Reviewer #1: Pros:

The authors do an excellent job of contextualizing the study providing clarity for non-archaeological reader.

The methodological approaches and analyses are presented in detail and conform to generally accepted procedures.

The conclusions as to the chronological dates for the log’s growth and cutting are securely established and generally supported by earlier research on the post’s chronological context and confirm the cultural context.

The Sr analyses are well conceived and support the conclusion that the post was obtained from a distant location to the south and justifies the interpretation that the inhabitants were engaged in the long-distance transport of such items.

Cons:

The interpretation of the timing and significance of the post is problematic. While the research confirms the dating of the post’s cutting, it does not relate to the post’s attempted extraction. Whether it relates to the cultural events of the site’s decline is at present hypothetical.

The post’s historical significance as a key interpretive event is further diminished by the authors’ account that hundreds of such posts likely dotted this landscape.

6. PLOS authors have the option to publish the peer review history of their article (what does this mean? ). If published, this will include your full peer review and any attached files.

**Do you want your identity to be public for this peer review?** For information about this choice, including consent withdrawal, please see our Privacy Policy .

Reviewer #1: No

---

## [Author Response · Author response to Decision Letter 1]

15 Sep 2025

We thank the editor and anonymous reviewer for their comments and suggestions. The critical point highlighted by the reviewer is an issue that the authors have considered carefully; prior to, during, and after writing. It is our view that a discussion of post extraction should remain in the article because it is of critical to contextualizing the historical importance of the post-pit phenomenon in the past. We have, in response to the concerns of the reviewer, revised our discussion of the timing of post extraction to emphasize that (1) we are not predicting an extraction date with the same resolution as our observed cutting date and that (2) the proposed timing of extraction is supported by independent information from other archaeological contexts and informs archaeological models for the process of the dissolution of the Cahokia phenomenon.

To address concerns about the hypothetical relationship between the proposed date of the poles extraction and its observed cutting date, we expand the explanation of the logic of our inferred indirect date for the pole extraction – that is based on the fact that when it was extracted the pole must have been weakened by rot at the ground surface sufficiently to break as it did. We re-phrase our discussion of this to clarify that this is an inference based on contextual clues from surrounding archaeological features and is copiously supported by references cited in the text. While not a formal hypothesis or model, this inference can be refuted with further research and is valid in the context of the discussion of the results. Importantly, we acknowledge the potential error inherent in such a claim and do not attempt to extend the precision of the pole’s cutting date to the inferred extraction date.

We also address the argument that the large number of marker posts on the ancient Cahokian landscape diminish the value of the post under study by clarifying in the introduction that the importance of dating marker posts is to gain a better understanding of the dissolution of Cahokian urbanism (a phenomenon of world historical importance) through charting the tempo of ritual marker pole emplacement. While we cannot do that with our sample of one, the paper’s contribution is that it clearly shows how this research could be accomplished technically. Furthermore, we emphasize that this is the first such marker pole, and a major and prominent one at that, to ever be directly dated at high-precision and sourced chemically.

---

## [Decision Letter · Decision Letter 1]

19 Sep 2025

Age and origin of a Cahokian Wooden Monument at the Mitchell Site, Illinois, USA

PONE-D-25-11477R1

Dear Dr. Kessler,

We’re pleased to inform you that your manuscript has been judged scientifically suitable for publication and will be formally accepted for publication once it meets all outstanding technical requirements.

Kind regards,

Peter F. Biehl, PhD

Academic Editor

PLOS ONE

Additional Editor Comments (optional):

Reviewer #1:

Reviewers' comments:

Reviewer's Responses to Questions

**Comments to the Author**

1. If the authors have adequately addressed your comments raised in a previous round of review and you feel that this manuscript is now acceptable for publication, you may indicate that here to bypass the “Comments to the Author” section, enter your conflict of interest statement in the “Confidential to Editor” section, and submit your "Accept" recommendation.

Reviewer #1: All comments have been addressed

2. Is the manuscript technically sound, and do the data support the conclusions?

Reviewer #1: Yes

3. Has the statistical analysis been performed appropriately and rigorously? 

Reviewer #1: Yes

4. Have the authors made all data underlying the findings in their manuscript fully available?

Reviewer #1: Yes

5. Is the manuscript presented in an intelligible fashion and written in standard English?

Reviewer #1: Yes

6. Review Comments to the Author

Reviewer #1: In the enhanced revised manuscript the authors have presented a clarified and well-reasoned contribution to the chronological and social history of Cahokia. It will be a contribution to regional history and more broadly to the development of urbanism in Native North America.

7. PLOS authors have the option to publish the peer review history of their article (what does this mean? ). If published, this will include your full peer review and any attached files.

**Do you want your identity to be public for this peer review?** For information about this choice, including consent withdrawal, please see our Privacy Policy .

Reviewer #1: No

---

## [Editor Report · Acceptance letter]

PONE-D-25-11477R1

PLOS ONE

Dear Dr. Kessler,

I'm pleased to inform you that your manuscript has been deemed suitable for publication in PLOS ONE. Congratulations! Your manuscript is now being handed over to our production team.

Kind regards,

on behalf of

Dr. Peter F. Biehl

Academic Editor

PLOS ONE